# Module-Aware Optimization for Auxiliary Learning

**Hong Chen,  Xin Wang**[*] **Yue Liu,  Yuwei Zhou,  Chaoyu Guan,  Wenwu Zhu**[*]

Tsinghua University

{h-chen20,liuyue17,zhou-yw21,guancy19}@mails.tsinghua.edu.cn

{xin_wang,wwzhu}@tsinghua.edu.cn

## Abstract

Auxiliary learning is a widely adopted practice in deep learning, which aims to improve the model performance on the primary task by exploiting the beneficial information in the auxiliary loss. Existing auxiliary learning methods only focus on balancing the auxiliary loss and the primary loss, ignoring the module-level auxiliary influence, i.e., an auxiliary loss will be beneficial for optimizing specific modules within the model but harmful to others, failing to make full use of auxiliary information. To tackle the problem, we propose a Module-Aware Optimization approach for Auxiliary Learning (MAOAL). The proposed approach considers the module-level influence through the learnable module-level auxiliary importance, i.e., the importance of each auxiliary loss to each module. Specifically, the proposed approach jointly optimizes the module-level auxiliary importance and the model parameters in a bi-level manner. In the lower optimization, the model parameters are optimized with the importance parameterized gradient, while in the upper optimization, the module-level auxiliary importance is updated with the implicit gradient from a small developing dataset. Extensive experiments show that our proposed MAOAL method consistently outperforms state-of-the-art baselines for different auxiliary losses on various datasets, demonstrating that our method can serve as a powerful generic tool for auxiliary learning[2].

## 1 Introduction

Auxiliary learning is a common practice in deep learning, which utilizes auxiliary losses (generally from related tasks) to benefit the primary task in terms of model performance or generalization ability. The auxiliary learning paradigm has shown its effectiveness in various areas including image classification [1, 2], recommendation [3–5], reinforcement learning [6, 7], etc. Under various scenarios, task-specific auxiliary losses are designed for different purposes, e.g., in recommendation, previous works [5, 3] add the click-through rate prediction loss to help the click conversion rate prediction; in image classification [1], the self-supervised auxiliary loss is designed to improve classification accuracy when labels are inadequate; and the losses of head pose estimation and facial attribute inference are utilized to aid facial landmark detection [8].

Existing auxiliary learning methods focus on balancing the importance of each auxiliary loss and the primary loss. Most of them assign a predefined weight to each auxiliary loss and then optimize the sum of the weighted auxiliary losses and the primary loss. The predefined weights generally need carefully tuning with Hyper-parameter Optimization (HPO) tools to prevent negative auxiliary transfer. Recently, several works [9, 10, 6] are proposed to automatically assign weights to each auxiliary loss based on the gradient similarity between each auxiliary loss and the primary loss. The more recent works [11] further proposes to utilize a non-linear auxiliary loss combination to better

---

[*]Corresponding Authors.

[2]Our code will be released at `https://github.com/forchchch/MAOAL`

exploit the beneficial auxiliary information, and [12] proposes to jointly select the most beneficial task and data for auxiliary learning.

Nevertheless, existing approaches ignore the important fact that an auxiliary loss could have different influences on different modules within the model. Here we define a module to be a fine-grained part of the model, e.g., a module can be a block in ResNet [13], a transformer block in Bert [14], or a layer in a Multi-Layer-Perceptron (MLP). The necessity of considering the module-level influence of auxiliary losses has been indicated by research related to multi-task learning. Previous multi-task learning works [15, 16] show that shallow layers (layers close to the input) contain more common information among related tasks, while deep layers (layers close to the output) contain more task-specific information that tends to cause negative transfer. This phenomenon indicates that when the auxiliary loss comes from a related task, if we do not consider the module-level influence, the negative information in the deep layers will counterbalance the benefits brought by the shallow layers, resulting in inefficient usage of auxiliary losses. However, these multi-task learning works [15, 16] aim at designing model architectures, not suitable for the auxiliary learning scenario where the model architecture is designed (and fixed) for the primary task, and our target is to utilize the auxiliary losses to better optimize the model. In sum, considering the module-level influence is important to auxiliary learning, and there has been no work on auxiliary learning considering this module-level influence.

To deal with the module-level auxiliary influence, we propose a module-aware optimization approach for auxiliary learning (MAOAL). The comparison of the existing methods and our module-aware optimization approach is shown in Figure 1. The proposed MAOAL approach considers the module-level influence of auxiliary losses by introducing the learnable module-level auxiliary importance, which reflects the importance of each auxiliary loss to each module within the model. Specifically, the module-level auxiliary importance and the model parameters are jointly optimized in a bi-level manner. In the lower optimization, the model parameters are updated with the module-level auxiliary importance parameterized gradient, so that each auxiliary loss can be used to optimize the module it is beneficial to, making each module of the model play a better role for the primary task. In the upper optimization, the module-level auxiliary importance is updated with the implicit gradient from a small developing dataset, making the module-level auxiliary importance more accurate. Additionally, the best-response approximation is utilized for efficient implicit gradient calculation. Since the module-level auxiliary importance is learned by a data-driven optimization process without requiring specific knowledge of the auxiliary task, the proposed module-aware optimization method can accommodate a variety of auxiliary losses. We conduct experiments on different tasks and datasets, and experimental results show that our method consistently outperforms state-of-the-art methods. Particularly, when we consider a wider range of auxiliary losses, such as adopting L2 regularization [17, 18] as an auxiliary loss, our method can also deal with the module-level influence in these situations and bring better model performance.

We summarize our contributions as follows. (i) We are the first to investigate the module-level influence in the general auxiliary learning setting. (ii) We propose a module-aware optimization (MAOAL) method for auxiliary learning, which utilizes the module-level auxiliary importance parameterized gradient to optimize the model parameters, and optimizes the module-level importance with the implicit gradient. (iii) Extensive experiments demonstrate that our method consistently outperforms state-of-the-art baselines in various scenarios. Analysis of the modules and the module-level auxiliary importance provides insights for auxiliary learning.

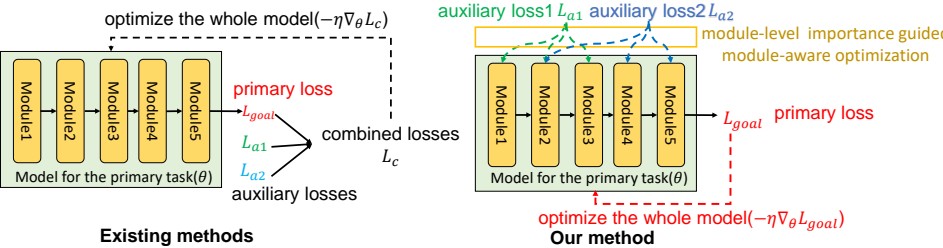

Figure 1: Existing methods utilize the combination of the primary loss and auxiliary losses to update the whole model parameters. In module-aware optimization, auxiliary loss 1 should update the parameters in module 1,2,3 and auxiliary loss 2 should update module 2,4 and 5.

## 2 Related Work

**Auxiliary Learning** Auxiliary learning is a common practice in deep learning. This learning paradigm aims to improve the model performance on the primary task with designed auxiliary losses (generally from auxiliary tasks). Most existing works [1, 3] utilize the auxiliary losses by linearly combining them with the primary loss and then utilizing the combined loss to optimize the whole model. The weights for the auxiliary losses will be tuned to avoid negative transfer to the primary task. Some recent works [9, 10, 6, 19] propose to automatically weight the auxiliary losses during the training process in a dynamic way. Specifically, work [9] calculates the gradient of each auxiliary loss, and utilizes the cosine similarity between the auxiliary gradient and that of the primary loss as the auxiliary loss weight. Work [10] shares a similar idea with [6], but with the goal that the weighted gradient sum should be close to the gradient of the primary task. Moreover, [11] proposes to learn a non-linear combination for auxiliary losses, and [12] proposes to select not only the task but also each data sample within each task to make full use of the auxiliary information. However, these methods only conduct model-level optimization for auxiliary losses, ignoring the fact that one auxiliary loss could contribute differently to different modules within the model.

**Multi-task learning** Another line of highly related work to optimize several losses is multi-task learning. Different from the goal of auxiliary learning, multi-task learning aims to obtain good performance on all the learned tasks, while learning with auxiliary losses only focuses on the primary task. Existing multi-task learning methods can be roughly categorized to three parts [20]: multi-task architecture design [21, 16], multi-task optimization [22–24] and multi-task relationship learning [25]. Multi-task architecture design methods aim at designing proper architectures for simultaneously learning multiple tasks. Some architecture design works [16, 21] point that different tasks share more common information in the shallow layers, which inspires us that module-aware optimization for auxiliary losses is needed. The multi-task optimization methods aim at optimizing the whole model for all tasks with methods like loss weighting [22, 23], which can be used to optimize multiple losses. Particularly, work [26] proposes to assign layer-wise to each loss, but it only focuses on multi-task setting and the weight is assigned based on heuristics, which cannot be directly applied to our general auxiliary setting where we only care about the performance of the primary task and the auxiliary losses might by some other regularization terms like L2 regularization or disentangled regularization [27–30].

**Position of this work** This work is mainly inspired by [31], where the machine is expected to automatically select the data, tasks and model to be learned. Previous works have focused on different aspects. [27, 32–35] focus on automatic data selection, [11, 6, 9] focus on automatic task selection, and [36–38] focus on automatic model designing. More recently, some works focus on joint selection of different elements, where the selection space is much more complicated. [12] proposes to joint select task and data, [39, 40] proposes to joint select data and model, and this work fills the joint task-model selection gap.

## 3 The Proposed Method

### 3.1 Preliminaries and Problem Formulation

Assume that we have a primary task $\mathbb{T}_{goal}$ and its corresponding loss $L_{goal}$ (primary loss). There are totally $K$ auxiliary losses $\{L_{a1}, L_{a2}, \cdots, L_{ai}, \cdots, L_{aK}\}$, utilized to help the primary task. Let $D_{train}$, $D_v$ respectively be the training and validation dataset. The model for the primary task is parameterized by $\theta$, which is a deep model composed of different modules, i.e., $\theta = [\theta^1, \theta^2, \cdots, \theta^j, \cdots \theta^m]$. The partition for the modules can be flexibly defined by users, e.g., a module can be a block in ResNet [13], a transformer block in Bert [14], or specified by its designed function [41]. Previous approaches for auxiliary learning generally adopt the following objective:

$$L_c(\theta) = L_{goal}(\theta; D_{train}) + \sum_{i=1}^{K} w_i * L_{ai}(\theta; D_{train}), \tag{1}$$

where $w_i$ is the weight used to balance the auxiliary losses and the primary loss. With proper values set for $w_i$, the model parameters $\theta$ are updated with the widely adopted gradient descent method where t is the unrolled step:

$$\theta_t = \theta_{t-1} - \eta \nabla_{\theta_{t-1}} L_c(\theta_{t-1}). \tag{2}$$

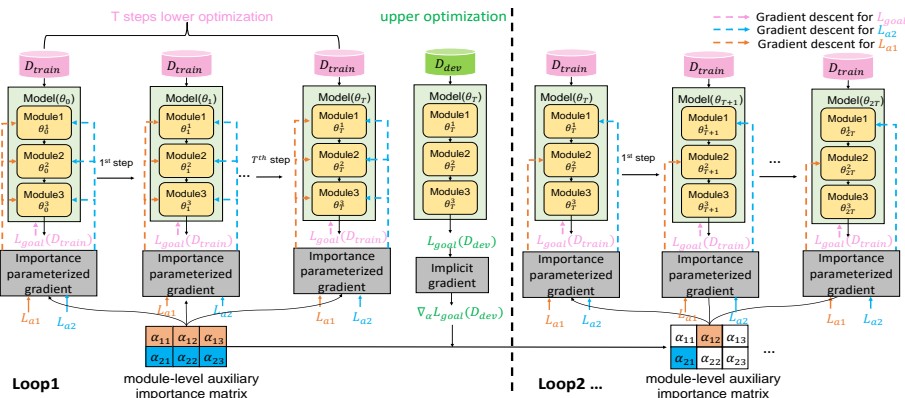

Figure 2: The module-aware optimization method for auxiliary learning. In each loop, the lower optimization is conducted for T times followed by one-step upper optimization. In the example, in Loop1, the two auxiliary losses $L_{a1}, L_{a2}$ update all modules equally. After Loop1 optimization, $L_{a1}$ is only important to module2 and is used for updating module2 while $L_{a2}$ only for module1.

Although this kind of learning paradigm has been widely used in various scenarios, it cannot deal with the module-level influence as indicated in [21, 16]. To avoid confusion, we remind that the superscript of $\theta^j$ indexes the module within the model, and the subscript of $\theta_t$ indexes the unrolled step in the rest of the paper.

Our goal is to deal with the module-level influence of auxiliary losses and make each auxiliary loss optimize the module which it is beneficial to. Specifically, we introduce the module-level auxiliary importance matrix $\alpha = \{\alpha_{ij}\}_{i=1}^{K}{}_{j=1}^{m}$, where $\alpha_{ij}$ is the importance of auxiliary loss $L_{ai}$ to the module $\theta^j$, and then optimize the model parameters $\theta$ for the primary task according to the module-level auxiliary importance. Here two key problems are how to obtain the optimal module-level auxiliary importance and how to utilize the importance to optimize the model parameters $\theta$. We optimize the module-level auxiliary importance and the model parameters in a bi-level manner, and the whole framework of the module-aware optimization process is shown in Figure 2. In the lower optimization, we optimize the model parameters $\theta$ with the importance parameterized gradient. In the upper optimization, we update the module-level auxiliary importance with the implicit gradient from a developing dataset $D_{dev}$. In the next two subsections, we describe the lower and upper optimization in detail and summarize the complete algorithm in Section 3.4.

### 3.2 Lower Optimization: Importance Parameterized Gradient for Model Parameters

In the lower optimization, we optimize the model parameters $\theta$ on the training dataset $D_{train}$. Assuming that we have already known the module-level auxiliary importance $\alpha_{ij}$, we will utilize the importance to guide the optimization process of the model parameters $\theta$. We first obtain the gradient of each module $\theta^j$ with respect to each auxiliary loss $L_{ai}$:

$$[\nabla_{\theta^1} L_{ai}, \nabla_{\theta^2} L_{ai}, \cdots, \nabla_{\theta^m} L_{ai}] = \nabla_\theta L_{ai}(\theta; D_{train}), i = 1, 2, \cdots, K \qquad (3)$$

Then we reweight the gradient of each auxiliary loss with the importance $\alpha_{ij}$ and obtain the importance parameterized gradient $\nabla_\theta \bar{L}(\theta, \alpha; D_{train})$ as follows:

$$\nabla_\theta \bar{L}_{ai}(\theta, \alpha_i; D_{train}) = [\alpha_{i1} \nabla_{\theta^1} L_{ai}, \alpha_{i2} \nabla_{\theta^2} L_{ai}, \cdots, \alpha_{im} \nabla_{\theta^m} L_{ai}], i = 1, 2, \cdots, K, \qquad (4)$$

$$\nabla_\theta \bar{L}(\theta, \alpha; D_{train}) = \nabla_\theta L_{goal}(\theta; D_{train}) + \sum_{i=1}^{K} \nabla_\theta \bar{L}_{ai}(\theta, \alpha_i; D_{train}), \qquad (5)$$

where in Eq.(4), the gradient for each auxiliary loss is masked with its module-level auxiliary importance, where if the $i^{th}$ auxiliary loss is important to optimizing the $j^{th}$ module, the auxiliary module-level gradient $\nabla_{\theta^j} L_{ai}$ will be weighted with a larger $\alpha_{ij}$ and vice versa. In Eq.(5), we combine all the weighted auxiliary gradient and the gradient of the primary loss to obtain the importance parameterized gradient, which will be finally used to optimize the model parameters $\theta$. After that, we utilize the widely adopted unrolled gradient descent to optimize the model parameters

$\theta$ as follows, where $\eta_1$ is the learning rate.

$$\theta_{t+1} = \theta_t - \eta_1 \nabla_{\theta_t} \bar{L}(\theta_t, \alpha; D_{train}) \tag{6}$$

## 3.3 Upper Optimization: Implicit Gradient for Module-level Auxiliary Importance

In the upper optimization, our target is to optimize the module-level auxiliary importance $\alpha_{ij}$. To obtain the gradient of $\alpha$, we introduce a small developing dataset $D_{dev}$ as in [42]. $D_{dev}$ is a small-size dataset for the primary task split from the validation dataset $D_v$. Since we expect that our model can finally perform well on the validation dataset $D_v$, our model should also perform well enough on its subset $D_{dev}$. Therefore, it is natural to utilize the loss on $D_{dev}$ to optimize the importance matrix $\alpha$. Assume that we have conducted M steps of lower optimization, and the current model parameter is $\theta_M$. The loss of the model on the developing dataset $L_{goal}(\theta_M(\alpha); D_{dev})$ can be calculated. Note that we only calculate the primary loss on $D_{dev}$ because we actually only care about the model performance on the primary task. Directly calculating the gradient of $\alpha$ with respect to $L_{goal}(\theta_M(\alpha); D_{dev})$ is not easy. The loss $L_{goal}(\theta_M(\alpha); D_{dev})$ directly relies on the model parameters $\theta_M$ instead of the importance matrix $\alpha$. $\alpha$ implicitly influences the value of $L_{goal}(\theta_M(\alpha); D_{dev})$ by influencing the gradient of $\theta$ as shown in the lower optimization in Eq.(6). According to this observation, we use the chain rule to obtain the the gradient of $\alpha$ with respect to $L_{goal}(\theta_M(\alpha); D_{dev})$:

$$\nabla_\alpha L_{goal}(\theta_M(\alpha); D_{dev}) = \nabla_{\theta_M} L_{goal}(\theta_M(\alpha); D_{dev}) \nabla_\alpha \theta_M \tag{7}$$

The term $\nabla_{\theta_M} L_{goal}(\theta_M(\alpha); D_{dev})$ is easy to obtain using the autograd tools. However, the term $\nabla_\alpha \theta_M$ requires more complex exploration to the lower unrolled steps. By differentiating and unrolling the process in Eq.(6), we obtain the following results:

$$\nabla_\alpha \theta_M = - \sum_{0 \le \tau < M} ( \prod_{0 \le j < \tau} [I - \eta_1 \nabla_\theta^2 \bar{L}(\theta, \alpha; D_{train})]|_{\theta_{M-1-j}}) \nabla_\alpha \nabla_\theta \eta_1 \bar{L}(\theta, \alpha; D_{train})|_{\theta_{M-1-\tau}}, \tag{8}$$

where $\tau$ indexes the steps we look back, summing up all the $M$ looking-back results gives the desired gradient. This means that if we know the Jacobi and the Hessian within the past $M$ steps, we can use them to calculate $\nabla_\alpha \theta_M$, the gradient of $\alpha$ with respect to current model parameters $\theta_M$. Detailed derivation can be found in the supplementary file. However, this kind of calculation is memory-consuming because we have to restore the Jacobi and Hessian during the past $M$ steps. To improve efficiency, we follow the best-response approximation in [42] and assume that $\theta_0, \theta_1, \cdots, \theta_{M-1}$ all equal to $\theta_M$. With this assumption, we can approximate Eq.(8) in the following manner:

$$\nabla_\alpha \theta_M \approx - \sum_{0 \le \tau < M} (I - \eta_1 \nabla_\theta^2 \bar{L}(\theta_M, \alpha; D_{train}))^\tau \nabla_\alpha \nabla_\theta \eta_1 \bar{L}(\theta_M, \alpha; D_{train}), \tag{9}$$

where we only use the Hessian and Jacobi at the current step. Replacing the term $\nabla_\alpha \theta_M$ in Eq.(7) with Eq.(9) gives the desired gradient of $\alpha$. Moreover, the calculation for Eq.(7) can be performed efficiently using vector-Jacobian products [42] as shown in Algorithm 1, which is both memory and computation efficient. This kind of approximation assumes that $\theta_\tau (0 \le \tau < M)$ equals to $\theta_M$, which will be more accurate when $\theta_\tau$ is around the best response $\theta^*$ under given $\alpha$(the best response $\theta^*$ is the point that the unrolled process in Eq.(6) converges to when $\alpha$ is fixed). This is why it is called best-response approximation in [42]. The effectiveness of this approximation is practically validated in both previous works [42, 43, 11] and our experiments.

## 3.4 Overall Algorithm

With both the lower and upper optimization above, we summarize the complete module-aware optimization algorithm for auxiliary learning in Algorithm 1. The input for the algorithm contains three datasets: the training dataset $D_{train}$, the developing dataset $D_{dev}$ and the validation dataset $D_v$, and the hyperparameters. $T$ and $\eta_1$ are used for the lower optimization, where $T$ is the steps we conduct lower optimization in one loop and $\eta_1$ is the learning rate for the lower optimization. $M$ and $\eta_2$ are used for the upper optimization, where M is the total looking-back steps in Eq.(9) and $\eta_2$ is the learning rate for the upper optimization. The model parameters are randomly initialized and the importance matrix is initialized with all its elements 1.0. Specifically, in line 4 to line 6, we conduct lower optimization for T times. In line 15 to line 22, we conduct the upper optimization, where line 16 to line 21 is the process that utilizes the vector-Jacobian products to calculate the gradient in

Eq.(7). The lower optimization and the upper optimization are iteratively conducted until converged. Particularly, in line 7 to line 14, we will evaluate the model performance on the validation dataset $D_v$ after each $interval$ loops of lower-and-upper optimization. We always record the $\theta$ with the best performance in $\theta_{opt}$, and finally return the parameters $\theta_{opt}$. The use of the validation dataset is to prevent the model from overfitting $D_{train}$ or $D_{dev}$. Note that we do not directly look back the total $T$ steps in the upper optimization, but only look back $M$ steps($M < T$) for faster computation of Eq.(7), which is also adopted by [42, 11].

In our algorithm, each auxiliary loss enjoys an independent vector $\alpha_i = \{\alpha_{ij}\}_{j=1}^m$, enabling it not only to balance losses, but also consider how important each auxiliary loss is to each module. Without requiring specific knowledge of auxiliary losses, the data-driven optimization of the module-level auxiliary importance enables our method to be applied to the general setting of auxiliary learning.

---

**Algorithm 1** Module-Aware Optimization for Auxiliary Learning (MAOAL)

---

1: **input:** dataset:$D_{train}, D_{dev}, D_v$; hyperparameters: $T, M, \eta_1, \eta_2, interval$
2: **initialize:** $\theta, \alpha, counter = 0, performance_{opt} = 0, \theta_{opt} = \theta$
3: **while** not converged **do**
4:     // lower optimization: update $\theta$ with fixed $\alpha$
5:     **for** $t = 0$ **to** $T - 1$ **do**
6:         Update model parameters $\theta$ using Eq.(6);
7:     **end for**
8:     // evaluate the model on $D_v$
9:     $counter = counter + 1$
10:     **if** $counter\%interval == 0$ **then**
11:         $performance = Metric(\theta; D_v)$;
12:         **if** $performance > performance_{opt}$ **then**
13:             $performance_{opt} = performance, \theta_{opt} = \theta$;
14:         **end if**
15:     **end if**
16:     // upper optimization: update $\alpha$ with current $\theta$ ;
17:     $p = v = \nabla_\theta L_{goal}(\theta; D_{dev})$;
18:     **for** $\tau = 1$ **to** M **do**
19:         $v \leftarrow v - grad(\eta_1 \nabla_\theta \bar{L}(\theta, \alpha; D_{train}), \theta, grad\_outputs = v)$
20:         $p \leftarrow p + v$
21:     **end for**
22:     $\nabla_\alpha L_{goal}(\theta; D_{dev}) = -grad(\eta_1 \nabla_\theta \bar{L}(\theta, \alpha; D_{train}), \alpha, grad\_outputs = p)$
23:     $\alpha \leftarrow \alpha - \eta_2 \nabla_\alpha L_{goal}(\theta; D_{dev})$
24: **end while**
25: **Return** $\theta_{opt}$

---

## 4 Experiments

In this section, we empirically assess the efficacy of our method on various tasks and datasets. Also, we provide ablations on how our method works and the module influence under different scenarios.

### 4.1 Experimental Setup

**Task and datasets (i)** We first focus on the image classification problem with auxiliary tasks. We conduct experiments on two fine-grained image classification datasets, CUB [44] and Oxford-IIIT Pet [45], and two widely adopted general image classification datasets, CIFAR10 and CIFAR100 [46]. Specifically, on the CUB dataset, there are total 200 species of birds and each image has the label of its attributes, like the wing color and bill shape. In general, it requires expert knowledge to classify the birds, but it is much easier to discriminate the attributes. Thus, it is a natural practice to regard bird species classification as the primary task, and use the wing color classification and bill shape classification losses as the auxiliary losses. On the Oxford-IIIT Pet, CIFAR10 and CIFAR100 dataset, we follow the practice in [1] and rotate each image with degrees of $\{0, 90, 180, 270\}$. The primary task is image classification and we add the rotation degree prediction loss as the auxiliary loss. Note

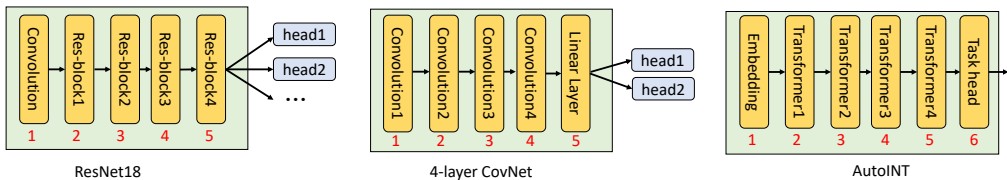

Figure 3: Adopted models and module partitions. Red number refers to module index.

that in the CUB datasets, we adopt pretrained backbone but in other datasets we train from scratch, to show the effectiveness of our method with different initialization. **(ii)** The second problem we focus on is the user rating prediction in recommendation. We evaluate our methods on two datasets with different sparsity, Amazon Beauty [47] and MovieLens1M [48]. Our primary task is to predict the rating for each user-item pair based on the user features and item features. In this problem, we consider a kind of more generalized auxiliary loss, L2 regularizer. In recommendation, L2 regularizer plays an important role to the model performance and is often used as an auxiliary loss [17, 18]. In this setting, the primary loss is the rating regression loss and the auxiliary loss is the L2 regularizer. Detailed dataset information can be found in the supplementary.

**Baselines** We compare our methods with different baselines that can be used for the general setting of auxiliary learning. Single loss learning (SLL) is a natural baseline that only utilizes the primary loss to train the model. Comparing with this baseline can help judge whether the auxiliary loss is beneficial to the primary task. Additionally, assigning each loss equal weights "1.0"(Equal) can be used to show the benefits of different auxiliary methods. Another natural baseline (HPO-tune) is to combine the auxiliary losses with predefined weights, and then use the HPO methods to tune the weights as conducted in [1]. GCS [9] is a baseline that utilizes the similarity between the gradient of each auxiliary loss and that of the primary loss to weight each auxiliary loss. AuxL [11] is a recently proposed method that learns a non-linear combination of all the losses. Uncert [22] is a method that is designed for combining the losses from multiple tasks, where the loss of each task is weighted according to their uncertainty. Note that for the methods that do not rely on $D_{dev}$, we add $D_{dev}$ to their training set for fair comparison.

**Implementation** We utilize the most widely adopted structure [8] for the tasks, with one common backbone followed by task-specific heads for all the tasks. In the image classification problem, the auxiliary losses come from some related auxiliary tasks and each task will have its corresponding task-specific head, but the recommendation problem only has one task head for the rating regression because its auxiliary loss is from L2 regularizer. For different tasks, we utilize different backbones. For the CUB and Pet dataset, we utilize ResNet18 [13] as the backbone. For the CIFAR10 and CIFAR100, we use a 4-layer convolutional network(ConvNet). For the rating prediction in Amazon-Beauty and MovieLens-1M, we choose AutoINT [49] as the backbone. The model structure and the module partition for the three settings are shown in Figure 3. We implement the task-specific heads with Multi-layer Perceptron(MLP) whose layer number is searched in $\{1, 2\}$. In ResNet18, the modules are divided by residual blocks, in 4-layer CovNet, the modules are divided by convolutional layer, and in AutoINT, the modules are divided by Transformer block. More detailed implementation can be found in the supplementary file.

## 4.2 Method Effectiveness

We evaluate our method on all the aforementioned tasks and datasets. For the image classification task, we utilize the top1-accuracy as the metric and for the rating prediction task, we utilize RMSE(rooted mean squared error) as the metric. Higher accuracy for image classification and lower RMSE for rating prediction indicate better performance. The overall results are shown in Table 1. As indicated in previous works [11, 9], losses from auxiliary tasks play a more important role when the labels for the primary task are inadequate. Therefore, we also conduct experiments on the CUB and Pet dataset, where for the primary task, we only utilize 20% labels. The results for this semi-supervised setting are shown in Table 2. We run all the experiments with three random seeds and report the mean and the std. Note that L2 regularizer with coefficient is too large for recommendation, so the Equal baseline performs worst on the Beauty and MovieLens dataset.

**Alleviating negative information from auxiliary loss** The results of CUB and MovieLens in Table 1 show that the auxiliary losses are harmful to the primary task. The baselines that utilize the auxiliary losses generally achieve worse performance than the SLL baseline that only utilizes the single primary loss to train. Our method shows its strong ability to alleviate the negative information from the auxiliary loss, it achieves comparative performance with the SLL baseline even though it does not use $D_{dev}$ for training. It is worth noting that the AuxL which also adopts a bi-level optimization framework shows excellent ability in alleviating the negative impact of the harmful auxiliary losses.

**Exploiting useful information in auxiliary losses** The results of Pet and Beauty in Table 1 and the results of Pet-semi and CUB-semi in Table 2 show that the auxiliary losses are beneficial to the primary task. All the baselines that utilize auxiliary losses show substantial improvement over the SLL baseline which only uses a single loss. Our method beats all the baselines in exploiting the beneficial auxiliary information, showing the advantage of the module-level consideration. The AuxL baseline fails to perform well enough probably because the form of the non-linear combination requires a specific design for the non-linear model in specific tasks.

**Other Observations (i)** Comparing the results of CUB and Pet in Table 1 to that in Table 2, we can see that when the labels are inadequate for the primary task, the auxiliary losses from related tasks will bring more benefits, which is consistent to previous works [11, 9]. **(ii)** The rotation prediction task seems to have little influence on the task of classification on the CIFAR10 and CIFAR100 dataset, the different methods that utilize the auxiliary rotation prediction loss do not bring substantial improvement or severely do harm to the task of CIFAR classification. This phenomenon may be caused by the low resolution of the CIFAR images, where the changes brought by rotation may not be so significant. **(iii)** Although the Beauty and the MovieLens experiments both utilize the L2 regularizer to help the rating prediction, the auxiliary regularizer shows different effects. In Beauty, the regularizer is helpful but in MovieLens, the regularizer is harmful. This phenomenon is easy to understand because the MovieLens dataset is dense while Beauty is sparse. Regularizer is important to the sparse Beauty by preventing the overfitting problem. In the dense MovieLens, data are enough for each user so the regularizer is not necessary.

Overall, when we utilize different kinds of auxiliary losses, we always expect to exploit the beneficial information while resisting the impact of negative information. Our proposed module-aware optimization method, shows its superiority when facing both beneficial and harmful auxiliary losses.

Additionally, based on the reviews, we add additional experiments on the NYUv2 [50] and CIFAR-100/20 datasets so that we can validate the ability of our method when facing more auxiliary tasks. In NYUv2, we totally have 3 tasks, i.e., segmentation, depth prediction and normal prediction, where the segmentation is the primary task and we adopt the EfficientNet [51] as the backbone, which is split into 8 blocks according to its model structure. In the CIFAR-100/20 dataset, we treated each of the 20 'coarse' classes as one task, where each coarse class contains 5 classes. We regard the "people" classification("baby", "boy", "girl", "man", "woman") as the primary task, and the 0-9 coarse class classification as the auxiliary tasks, totally 11 tasks. The ResNet18 is adopted as the backbone for the CIFAR-100/20. The results of the added experiments are shown in Table 3. These experiments show that our method can comply well with more auxiliary losses and also more advanced models.

Table 1: Performance of different methods under different scenarios. The methods with the best performance are bolded.

| | Dataset | | | | | |
|---|---|---|---|---|---|---|
| **Method** | CUB | Pet | CIFAR10 | CIFAR100 | Beauty | MovieLens |
| | ACC(%) | ACC(%) | ACC(%) | ACC(%) | RMSE | RMSE |
| SLL | $77.29_{0.63}$ | $61.14_{1.03}$ | $71.60_{0.25}$ | $46.52_{0.42}$ | $1.1008_{0.0006}$ | $0.9068_{0.0028}$ |
| Equal | $71.68_{0.87}$ | $67.52_{0.58}$ | $70.60_{0.30}$ | $45.57_{0.25}$ | $2.4512_{0.0013}$ | $2.2567_{0.0052}$ |
| HPO-tune | $76.72_{0.71}$ | $66.91_{0.46}$ | $71.51_{0.05}$ | $46.33_{0.24}$ | $1.0918_{0.0004}$ | $0.9379_{0.0008}$ |
| GCS | $71.59_{0.10}$ | $66.69_{0.77}$ | $70.95_{0.02}$ | $45.69_{0.01}$ | $1.0956_{0.0018}$ | $0.9137_{0.0001}$ |
| Uncert | $71.64_{1.15}$ | $67.42_{0.81}$ | $71.00_{0.42}$ | $46.37_{0.01}$ | $1.0942_{0.0004}$ | $0.9883_{0.0001}$ |
| AuxL | $76.99_{0.81}$ | $66.41_{0.97}$ | $71.58_{0.05}$ | $46.98_{0.06}$ | $1.0893_{0.0028}$ | $0.9063_{0.0018}$ |
| MAOAL | $\mathbf{77.30}_{0.17}$ | $\mathbf{69.61}_{0.93}$ | $\mathbf{72.24}_{0.27}$ | $\mathbf{47.57}_{0.40}$ | $\mathbf{1.0873}_{0.0017}$ | $\mathbf{0.9045}_{0.0013}$ |

Table 2: Performance of different methods with only 20% label on the primary task.

| Dataset | SLL | Equal | HPO-tune | GCS | Uncert | AuxL | MAOAL |
|---------|-----|-------|----------|-----|--------|------|-------|
| **CUB-semi** | $48.15_{0.88}$ | $50.54_{0.55}$ | $50.34_{0.71}$ | $51.06_{0.59}$ | $50.63_{0.16}$ | $48.50_{0.74}$ | $\mathbf{52.36_{0.64}}$ |
| **Pet-semi** | $31.81_{1.17}$ | $49.07_{0.94}$ | $49.28_{1.25}$ | $47.16_{0.57}$ | $47.96_{1.04}$ | $46.10_{0.58}$ | $\mathbf{51.42_{0.31}}$ |

Table 3: Performance of different methods in the setting with more auxiliary tasks.

| NYUv2 | SLL | Equal | HPO-tune | Uncert | GCS | AuxL | MAOAL |
|-------|-----|-------|----------|--------|-----|------|-------|
| m-IOU(%) | $32.61_{0.21}$ | $33.15_{0.15}$ | $33.09_{0.25}$ | $32.93_{0.32}$ | $32.18_{0.34}$ | $33.06_{0.14}$ | $\mathbf{33.88_{0.23}}$ |
| pixel-acc(%) | $68.42_{0.62}$ | $68.11_{0.24}$ | $68.52_{0.52}$ | $68.74_{0.62}$ | $67.58_{0.80}$ | $68.36_{0.53}$ | $\mathbf{69.17_{0.44}}$ |
| **CIFAR-100/20** | SLL | Equal | HPO-tune | Uncert | GCS | AuxL | MAOAL |
| acc(%) | $49.73_{0.61}$ | $51.27_{0.61}$ | $51.75_{0.72}$ | $48.40_{0.72}$ | $51.80_{0.40}$ | $51.53_{0.23}$ | $\mathbf{53.87_{0.46}}$ |

## 4.3 Module-level Auxiliary Importance

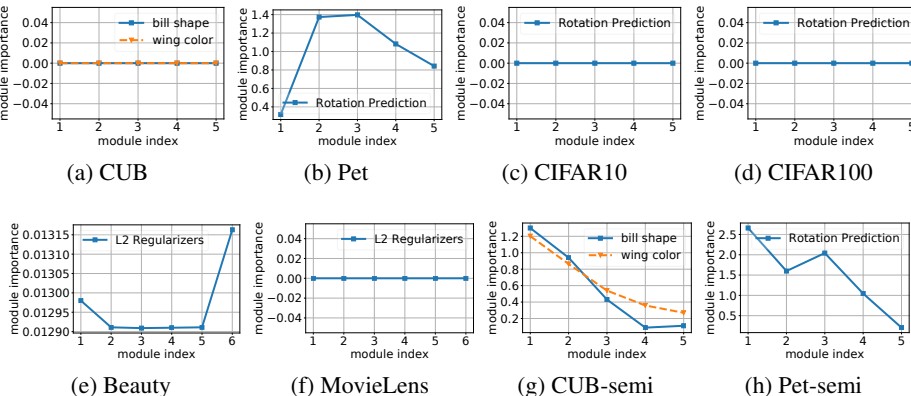

Figure 4: Module-level auxiliary importance

We expect that the learned module-level auxiliary importance can offer us some insights for auxiliary learning. The learned module importance under each scenario is shown in Figure 4 and the modules are numbered same as in Figure 3. **(i)** When the auxiliary losses are harmful (CUB,MovieLens) or have no substantial influence(CIFAR10,CIFAR100) to primary task, the final learned module-level auxiliary importance will be 0. It is a little surprising to note that in these harmful or unrelated auxiliary settings, our method can achieve comparative or even better performance than the SLL (only use single primary loss) baseline even though we utilize fewer data to train the model. This improvement may benefit from the dynamic weighting and training process. Although the weights for the auxiliary losses are finally set to zero, in the training process, the bi-level framework can utilize the gradient from the auxiliary losses to help the model generalize better on the unseen small $D_{dev}$. That is to say, the gradient from the auxiliary losses may guide $\theta$ to a better descent direction for the primary task in the early stages, but the gradient will show little importance when the better direction is found. **(ii)** When the auxiliary losses come from some related beneficial tasks (CUB-semi, Pet-semi), the importance of modules fits our intuition where the modules close to the input show higher importance while the modules close to the output show lower importance, and this is why our method can better exploit beneficial auxiliary information. Our module-aware optimization stops the auxiliary losses from optimizing deep modules (close to output), thus avoiding negative transfer in deep modules. **(iii)** However, in (b)Pet, module 1 shows low importance although the module 2,3,4,5 still follows the previous pattern, probably because compared to Pet-semi, the labels in Pet for the primary task are enough, so the information from the primary loss is enough to capture the low-level feature in the first module, and the gradient from the rotation loss is not so necessary. **(iv)** L2 regularizer is beneficial to the Beauty dataset, but its pattern is also different from that of CUB-semi and Pet-semi. Module6 (task head) and module1 (embedding table) need larger L2 regularizer than middle transformer blocks.

The learned module-level auxiliary importance shows different patterns under different scenarios, indicating our method can be applied to a variety of auxiliary learning scenarios.

### 4.4 Influence of Module Choice

In this part, we explore the influence of how we choose the module. We choose different partitions of modules and conduct experiments on the CUB-semi and Pet dataset. The results are shown in Table 4. The Parameter-L means we regard each parameter in the network as a module in our algorithm, which is a finer-grained partition. The Block-L means we regard each block in the ResNet as a module(the same as all the previous experiments). The Model-L means we regard the whole model as a module, which degenerates to the loss weighting methods. We can see that the Model-L variant performs worse than the Block-L, indicating the importance of considering the module-level effect of auxiliary losses. Additionally, we find that regarding each parameter as a module does not bring substantial improvement compared to the Block-L. This may be due to the reason that such a fine-grained module partition will make the importance matrix $\alpha$ contain more parameters. Optimizing more parameters in the upper level requires more data samples in $D_{dev}$ [42], which prevents the finer-grained partition from bringing more improvement. We also find that the performance of Paramter-L has obvious performance drop compared to Block-L in the Pet experiment, while in the CUB-semi experiment, the performance drop is less. This is likely because in the CUB-semi setting, we adopt the pretrained backbone for finetuning which can prevent overfitting problem to some extent, but in the Pet experiment we train from scratch.

Table 4: Performance of different choice of modules.

| Dataset | Parameter-L | Block-L | Model-L |
|---------|-------------|---------|---------|
| **CUB-semi** | $52.48_{0.79}$ | $52.36_{0.64}$ | $51.16_{0.42}$ |
| **Pet** | $66.14_{0.60}$ | $69.61_{0.93}$ | $67.18_{0.75}$ |

### 4.5 Module Complexity Influence on Importance with L2 Regularizer

We also explore how the module complexity will influence the learned importance when the auxiliary loss is the L2 regularizer. We change the hidden dimensions of the task head (2-layer MLP) of AutoINT in the Beauty dataset, and plot the learned importance for each module in Figure 5. Specifically, we change the hidden dimension of module6 from 8 to 128, and plot their auxiliary module-level importance. It can be seen that small hidden dimensions like hidden8 and hidden16 will finally bring a small regularizer for this module while larger hidden dimensions like hidden128 will bring comparatively larger importance. This fits our experience that the higher module complexity requires a larger regularizer to prevent overfitting.

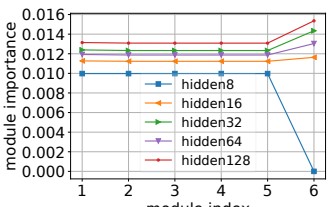

Figure 5: Importance changes with module hidden dimension

## 5 Conclusion

In this paper, we propose a module-aware optimization approach for the commonly encountered auxiliary learning scenario. Our method fills the gap that there exists module-level influence of auxiliary losses but there is no general method that can deal with this effect. We conduct extensive experiments to show that our proposed method can achieve superior performance with auxiliary losses from different scenarios, which could serve as a powerful tool when we want to obtain benefits from the auxiliary losses. Our method has no negative social impact. Despite its effectiveness, the only limitation of our method could be the computational cost is almost the same as that of the GCS baseline, i.e., we should calculate the gradient for each auxiliary loss and the training time is linear to the number of auxiliary losses. Reducing this kind of additional cost is an interesting future direction.

## Acknowledgement

This work was supported in part by the National Key Research and Development Program of China No. 2020AAA0106300 and National Natural Science Foundation of China (No. 62250008, 62222209, 62102222).

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
