# Module-Aware Optimization for Auxiliary Learning

**Hong Chen, Xin Wang,**$^*$ **Yue Liu, Yuwei Zhou, Chaoyu Guan, Wenwu Zhu**$^*$
Tsinghua University
{h-chen20,liuyue17,zhou-yw21,guancy19}@mails.tsinghua.edu.cn
{xin_wang,wwzhu}@tsinghua.edu.cn

## 1  Derivation of the Implicit Gradient

To obtain the gradient of $\alpha$, we utilize the loss on the developing dataset $L_{goal}(\theta_M; D_{dev})$, where $\theta_M$ gets connected to $\alpha$ through the following unrolled step:

$$\theta_M = \theta_{M-1} - \eta_1 \nabla_{\theta_{M-1}} \bar{L}(\alpha, \theta_{M-1}; D_{train}) \tag{A1}$$

Therefore, by unrolling and differentiating (A1) we can obtain the following process, where for simplicity we denote the loss on $D_{train}$ as $L_t$, and loss on $D_{dev}$ as $L_{dev}$.

$$\nabla_\alpha \theta_M = \nabla_\alpha \theta_{M-1} - \eta_1 \big[ \nabla_\alpha \nabla_{\theta_{M-1}} \bar{L}_t(\alpha, \theta_{M-1}) + \nabla^2_{\theta_{M-1}} \bar{L}_t(\alpha, \theta_{M-1}) \nabla_\alpha \theta_{M-1} \big] \tag{A2}$$

$$= (I - \eta_1 \nabla^2_{\theta_{M-1}} \bar{L}_t(\alpha, \theta_{M-1})) \nabla_\alpha \theta_{M-1} - \eta_1 \nabla_\alpha \nabla_{\theta_{M-1}} \bar{L}_t(\alpha, \theta_{M-1})$$

$$= (I - \eta_1 \nabla^2_{\theta_{M-1}} \bar{L}_t) \big[ (I - \eta_1 \nabla^2_{\theta_{M-2}} \bar{L}_t) \nabla_\alpha \theta_{M-2} - \eta_1 \nabla_\alpha \nabla_{\theta_{M-2}} \bar{L}_t \big] - \eta_1 \nabla_\alpha \nabla_{\theta_{M-1}} \bar{L}_t$$

$$= \cdots$$

$$= - \sum_{0 \le \tau < M} \big( \prod_{0 \le j < \tau} [I - \eta_1 \nabla^2_{\theta_{M-1-j}} \bar{L}_t] \big) \nabla_\alpha \nabla_{\theta_{M-1-\tau}} \eta_1 \bar{L}_t$$

In line 2 of (A2), we obtain the relation between $\nabla_\alpha \theta_M$ and its previous term $\nabla_\alpha \theta_{M-1}$, like a linear function with its weight $(I - \eta_1 \nabla^2_{\theta_{M-1}} \bar{L}_t(\alpha, \theta_{M-1}))$ and bias term $-\eta_1 \nabla_\alpha \nabla_{\theta_{M-1}} \bar{L}_t(\alpha, \theta_{M-1})$ . In the following derivation, $\nabla_\alpha \theta_{M-1}$ can be replaced with its previous linear form of $\nabla_\alpha \theta_{M-2}$, until we reach the initial $\nabla_\alpha \theta_0$, which equals to $0$. Summing up all the bias terms gives the final expression.

From (A2), the derivation of $\nabla_\alpha \theta_M$ requires the Jacobi and Hessian matrix in previous M steps, which is memory consuming. By approximating all previous $\theta_\tau$ as $\theta_M$, we obtain the following best-response approximation:

$$\nabla_\alpha \theta_M \approx - \sum_{0 \le \tau < M} (I - \eta_1 \nabla^2_{\theta_M} \bar{L}_t)^\tau \nabla_\alpha \nabla_{\theta_M} \eta_1 \bar{L}_t, \tag{A3}$$

Finally, we can obtain the gradient of $\alpha$ with respect to $L_{dev}$ using the chain rule.

$$\nabla_\alpha L_{dev}(\theta_M) = \nabla_{\theta_M} L_{dev}(\theta_M) \nabla_\alpha \theta_M \tag{A4}$$

Utilizing (A4), we can obtain the desired implicit gradient with the appproximated $\nabla_\alpha \theta_M$.

## 2  Experimental Details

We conduct experiments on the CUB [1], Oxford-IIIT Pet [2], CIFAR10 [3], CIFAR100, Amazon Beauty [4], and MovieLens1M [5], 6 datasets in total. The detailed training, validation, test and developing split is shown in Table 1, where $|D|$ denotes the number of samples in dataset D. We

---

$^*$Corresponding Authors.

36th Conference on Neural Information Processing Systems (NeurIPS 2022).

Table 1: Dataset statistics

|  | Beauty | MovieLens-1M | CUB | Pet | CIFAR10 | CIFAR100 |
|---|---|---|---|---|---|---|
| $|D_{train}|$ | 140,491 | 590,697 | 5,974 | 3,606 | 24,950 | 49,950 |
| $|D_{dev}|$ | 512 | 512 | 20 | 74 | 50 | 50 |
| $|D_v|$ | 17,625 | 73,901 | 2,897 | 1,835 | 25,000 | 5,000 |
| $|D_{test}|$ | 17,626 | 73,902 | 2,897 | 1,834 | 10,000 | 5,000 |

first split the dataset to training $D_{t,total}$, validation $D_v$ and test dataset $D_{test}$, following the common practice. Then we randomly sampling $|D_{dev}|$ samples from the $D_{t,total}$, and the $D_{t,total}$ is divided into the presented $D_{train}$ and $D_{dev}$. For the methods that do not rely on $D_{dev}$ for training, we use $D_{t,total}$ as their training set, while in our method, we only use $D_{train}$ as training set. Note that the size of $D_{dev}$ is much smaller than that of $D_{train}$. Then we describe in detail how we conduct experiments on each dataset.

## 2.1 CUB Experiments

**Dataset** On the CUB dataset, there are totally 11788 images of 200 species of birds. Each image has labels for the attributes of the bird. We regard the bird classification as the primary task, and the attribute classification as the auxiliary task. Specifically, we choose two attributes, one is the wing color of the bird, and the other is the bird bill shape. We follow the literature to crop all the images to size 256 [6]. During training, the cropped images will be randomly cropped to 224 followed by horizontally flip and Z-score normalization. During the test, the 256-size images are center-cropped to 224 followed by Z-score normalization.

**Details** We adopt the pretrained ResNet18 [7] and finetune it, assigning a one-layer head for each task. To train the model, we adopt SGD optimizer with learning rate 0.01 and momentum 0.0 for both our method and the baselines, the batchsize is 32 and we totally train 100 epochs. The hyper-parameter for the upper optimization in our method is set as follows: the optimizer is Adam with learning rate 1e-2, the lower optimization step T is 5 and the looking-back step M is 3 (the looking-back step is fixed for the rest experiments and we find it is effective enough to handle various scenarios). For the semi-supervised setting, we sample 20% data for each specie of birds for $D_{train}$, the batchsize is adjusted to 64 and the learning rate is adjusted to 5e-3 searched by the SLL baseline. The lower optimization step for the semi-supervised setting is 3.

## 2.2 Oxford-III Pet Experiments

**Dataset** On the Pet dataset, there are totally 7349 images of 37 species of pets. Each image is rotated with angle {0, 90, 180, 270}. The primary task is the pet classification and the auxiliary task is to predict the rotation angle. The image preprocessing is the same as the CUB dataset.

**Details** Different from fintuning in CUB experiments, we also want to verify whether our method performs well when training from scratch. We adopt ResNet18 without pretraining and assign a one-layer head for each task. To train the ResNet18 based model, we adopt Adam optimizer with learning rate 1e-4 for both our method and the baselines, the batchsize is 64 and we totally train 200 epochs. The hyper-parameter for the upper optimization in our method is set as follows: the optimizer is Adam with learning rate 1e-2, the lower optimization step T is 50. For the semi-supervised setting, we sample 20% data for each specie of pets for $D_{train}$ and the other hyper-parameters are the same. Additionally, for the Parameter-L variant, we regard each item in (model.named_parameters() in PyTorch) as a module.

## 2.3 CIFAR Experiments

**Dataset** On the CIFAR10 and CIFAR100 dataset, there are respectively 10 and 100 categories of images. Each image is rotated with angle {0, 90, 180, 270}. The primary task is the general image classification and the auxiliary task is to predict the rotation angle. During training, the images are randomly cropped to size 32 with padding 4, and then are normalized with the Z-score. During test, the images are directly normalized with the Z-score.

**Details** We adopt a 4-layer CovNet as the backbone. Each layer is composed of the {Conv2d, BatchNormalization, ReLU} components. The number of output channels for all the layers is set to 32, and the kernel size for the Conv2d is $3 \times 3$ and the stride is 1. After the 2nd and the 4th layer, there is a $2 \times 2$ Maxpooling layer. The head for each task is a linear model. We adopt the SGD optimizer with learning rate 0.01, momentum 0.9, and cosine annealing scheduler to train the model. The batchsize is 256 and we train 200 epochs for all the methods. The hyper-parameters for the upper optimization are as follows: the lower step T is 20 and the optimizer is Adam with learning rate 1e-2.

## 2.4 Amazon Beauty& MovieLens-1M Experiments

**Dataset** Based on the features of each user and item, we predict the rating of each user towards an item. In the Amazon Beauty dataset, the used feature contains the user ID, item ID and item category. In the MovieLens-1M dataset, the used feature contains user ID, user gender, user occupation, user age, item ID and item category. L2 regulaizer is added as the auxiliary loss to help the rating prediction. The sparsity for Beauty dataset is 0.0132%, and the sparsity for the MovieLens-1M dataset is 4.78%, where the sparsity means ratio of the number of the total interactions to the product of the user numbers and item numbers.

**Details** The adopted backbone for the experiments is AutoINT [8], which is composed of 4 transformer encoder blocks with head number 4 and embedding dimension 16. The rating prediction head is a 2-layer MLP with hidden dimension 16. To train the model, we adopt Adam optimizer with learning rate 1e-3 and batchsize 256. The lower optimization step T is 20. We totally train 20 epochs for all methods. For the upper optimization, we adopt the SGD optimizer with learning rate 0.001 and momentum 0.9 for MovieLens, and learning rate 0.1 with momentum 0.0 for Amazon Beauty.

## 2.5 Experimental Platform

We implement our method with PyTorch, and our experimental environments are as follows:

- CPU:Intel(R) Xeon(R) CPU E5-2699 v4 @ 2.20GHz

- RAM: 1TB DDR4

- GPU: 8x GeForce GTX 3090 Ti

- Operating System: Ubuntu 18.04.1 LTS

- Tools: Python3.7.0, PyTorch 1.4.0.

# 3 Weights Evolution During Training

We visualize the changes of the module-level auxiliary importance weights during training, and the results for each of the experiments are shown in the following subsections. For clear presentation, in all the datasets, we uniformly split the training process(from beginning to achieving the best validation performance) into fixed number of stages, and plot the importance weight of each loss to each module at the end of each stage.

## 3.1 CUB Experiments

**CUB** It can be seen from Figure 1 that the Bill shape and Wing color weights will finally become zero. In the early stages, the bill shape task is more beneficial than the wing color task, and the auxiliary losses are most beneficial to the shallowest module(module 0).

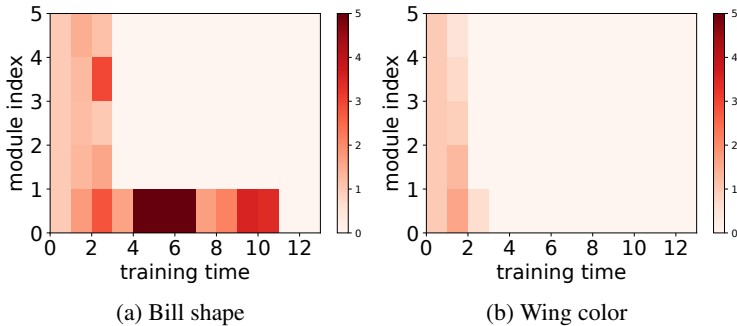

(a) Bill shape  (b) Wing color

Figure 1: Module-level auxiliary importance evolution during training: CUB

**CUB-semi** It can be seen from Figure 2 that when the labels for the primary task are inadequate, the bill shape and wing color auxiliary losses are both important to the primary task. As the training goes on, the importance weight of the deep modules(close to the output) will decrease, and the losses will have higher importance to the shallower modules(module 0 and 1).

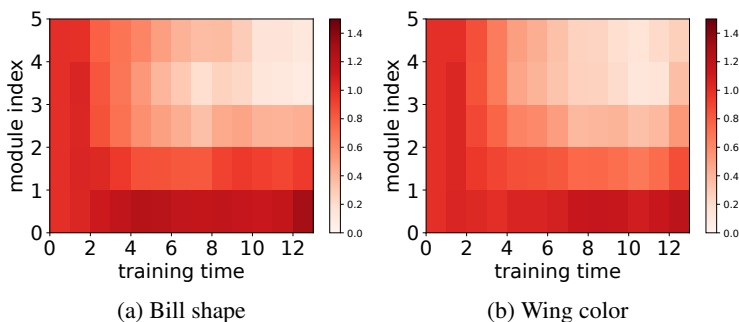

(a) Bill shape  (b) Wing color

Figure 2: Module-level auxiliary importance evolution during training: CUB-semi

### 3.2 Pet Experiments

**Pet** It can be seen from Figure 3(a) that the importance of the rotation loss in the shallowest module(module 0) will gradually decrease, while the rotation loss plays a more and more important role to module 1 and 2, which is different from that of the CUB-semi setting as analyzed in our paper.

**Pet-semi** It can be seen from Figure 3(b) that the importance of the rotation loss in the shallow modules(module 0,1,2) will gradually increase, while its importance to the last module will gradually decrease. Compared to Figure 3(a), we find that the number of labels for the primary task will influence the module-level auxiliary importance even the tasks are the same.

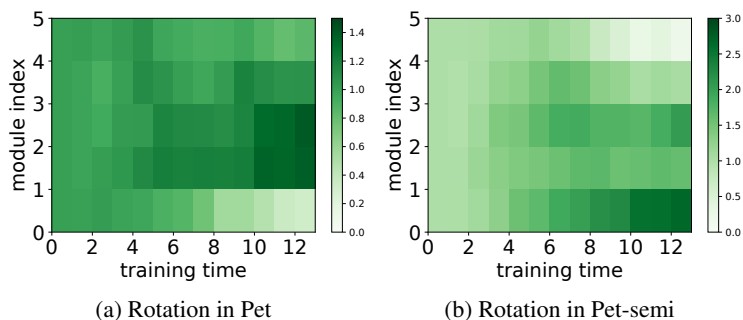

(a) Rotation in Pet  (b) Rotation in Pet-semi

Figure 3: Module-level auxiliary importance evolution during training: Pet

### 3.3 CIFAR Experiments

**CIFAR10 and CIFAR100** We can see from Figure 4 that the importance of the rotation loss to all the modules will become zero in the late stages. And interestingly, the behavior in CIFAR10 and CIFAR100 is quite similar: the importance of shallow modules(module 0 and 1) will become zero firstly, and gradually to the deep modules, until the importance of all modules to zero.

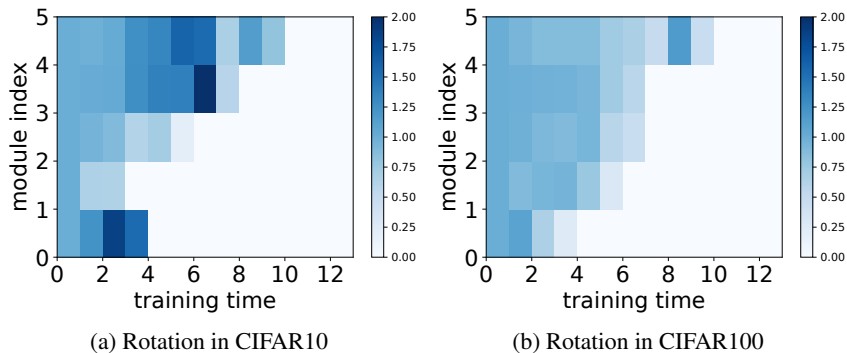

(a) Rotation in CIFAR10        (b) Rotation in CIFAR100

Figure 4: Module-level auxiliary importance evolution during training: CIFAR

### 3.4 Recommendation Experiments

**Amazon Beauty** As shown in Figure 5(a), the importance of the regularizer to each module is increasing during training, and the last module finally deserves the largest regularizer.

**MovieLens** As shown in Figure 5(b), the importance of the regularizer to each module is gradually decreasing to zero, because the data in MovieLens is rich, and there is no need for regularizer to prevent overfitting.

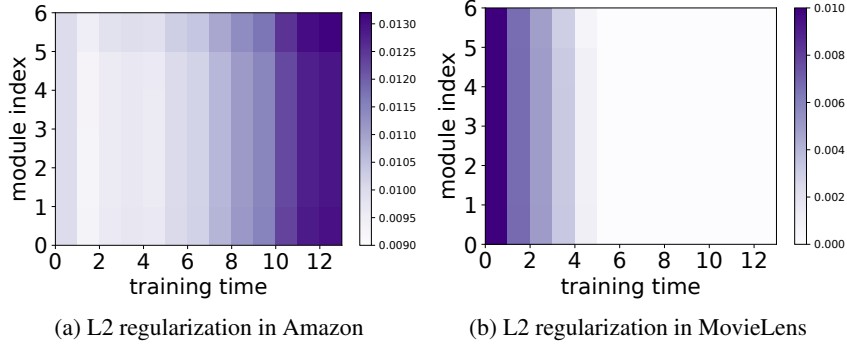

(a) L2 regularization in Amazon        (b) L2 regularization in MovieLens

Figure 5: Module-level auxiliary importance evolution during training: Recommendation

## 4 Time Complexity Analysis

Regarding the time of joint optimization of multiple losses with fixed weights as unit "1" and considering that the computational differences of different methods mainly come from the backward process, the backward time complexity of different methods is as follows:

- SLL(single loss learning): $O(1)$.
- HPO-tune: $O(R)$, in our method we use random search for the weights, R is the maximal trial numbers of the random search process.
- uncert: $O(1)$

- GCS(gradient cosine similarity): O(N), N is the number of losses. Since GCS will calculate the gradient for each loss, it will need N times of backward when updating the model parameters.

- AuxL: O(1+5/T). This method is a bi-level approach, we compare its time complexity with the fixed weight training method in one lower-upper loop. In one loop, the model will conduct T times of lower optimization and 1 upper optimization. The cost time for fixed weight training will be O(T). AuxL will have the additional uppper optimization which requires (M+2) for the Jacobi prediction where M is the looking-back steps, so it needs total (T+M+2) backwards, which results in O((M+T+2)/T) complexity compared to the unit. Since M in our experiments is fixed to 3, AuxL needs about O(1+5/T) complexity.

- MAOAL: O(N+N/T). Compared to AuxL, our method needs to calculate the gradient for each loss in the lower optimization, which requires TN backward. In the upper optimization, our method has (N-1) more backward than AuxL to calculate the gradient of each loss. Therefore, the time complexity is O((TN+N+M+1)/T) = O(N+N/T) in our experiment.