# OpenReview forum: "Module-Aware Optimization for Auxiliary Learning"
_NeurIPS.cc/2022/Conference — NeurIPS 2022 Accept_

### Official Review · Reviewer_xStN · 2022-07-04

**Rating:** 5
**Confidence:** 4
**Soundness:** 3 good
**Presentation:** 2 fair
**Contribution:** 1 poor

**Summary:**

Traditional approaches to auxiliary learning learn a single scalar per-task to adapt auxiliary tasks to the primary task.
This paper suggests to learn a scalar per-module for each auxiliary task. The motivation behind this is to allow more fine-grained adaptability in tuning the auxiliary task to improve the primary task. The authors show through experiments that this is an improvement over past methods they considered.

**Questions:**

1. Could you state the criteria you used to bold the numbers in your tables ? If it is highest average, please state this clearly. Some of the bolded numbers are *not statistically significant* but are still bolded.

2. Were all the table 1, 2 experiments conducted with Block-L ?

3. Is there any principled way / recommendation for deciding when to use Parameter/Block/Model-L ?

4. How do the results change with model size ? With the size of the dev set ?

5. It seems that for each experiment/data all methods only used a single learning rate ? It would be worth strengthening all methods by increasing the space of hyper-parameters (especially learning rate) searched over


Less of a question and more of a note - It would be interesting to see the evolution of the module level weights over time instead of just at the end of training as provided in Figure 4 .

**Limitations:**

This work introduces multiple hyper-parameters which may have to be cross-validation
1. The granularity of task adaptivity
2. M - the number of look-back steps

Computing the meta-gradient adds extra overhead but the runtime compared to simple fixed unitary weighting is not discussed

**Strengths And Weaknesses:**

**Strengths**

1. The paper is well written and easy to parse
2. The motivations for the work we clearly stated and situated in the context of previous work
3. Experimental evaluation over several datasets and multiple baselines over multiple seeds with calculated standard deviations.

**Weaknesses**

My primary concern with this paper is that the method seems incremental. Particularly,  all the core elements in this work can be stitched together from previous work (which itself would not be criticism per-se if there was novelty in the way the stitching together was done).
Specifically, learning auxiliary task adaptive weights via bi-level optimization has been done in [1, 2].
For adapting auxiliary tasks weights on a per-module bases, there are multiple related works that are not referenced.  Layer-adaptive weights are not uncommon in the multitask literature see Section 3.2 of [3] which introduces an index k, for each layer.
Also note that in [4], they use the primary task to learn a subspace within which they deconstruct the auxiliary task weights - this abstracts away the choice of Parameter/Block/Model-L introduced in this paper - and represents an automated "modularization" approach


**Relevant Related Work**

[1] Should we be pre-training? an argument for end-task aware training as an alternative : https://arxiv.org/abs/2109.07437

[2] Auxiliary learning by implicit differentiation:  https://arxiv.org/abs/2007.02693

[3] Gradient Vaccine: Investigating and Improving Multi-task Optimization in Massively Multilingual Models. https://arxiv.org/abs/2010.05874

[4] Auxiliary task update decomposition: the good, the bad and the neutral. https://arxiv.org/abs/2108.11346

---

> ### Author Response · Authors · 2022-08-01
> **Reply to the weakness part of Reviewer xStN**
>
> Thank the reviewer for the constructive comments. We address your concerns point by point as follows.
>
> 1.  **Reply to the Weakness**:
>
>     **Novelty of this work**: The reviewer considers that our proposed method is incremental compared to the given 4 related works. However, our work has clear innovations compared to the 4 references. We detail the differences and novelty as follows:
>
>     + [1][2] adopt the bi-level optimization method for auxiliary learning. However, our work has innovation compared to the two works in two aspects:
>
>         + Module level impact consideration: [1][2] only adopts the bi-level optimization method to balance the weights among losses, but our work further considers balancing the impact of different losses on different modules.
>
>         + Bi-level optimization method: Denoting the model parameters as $\theta$, the loss weights to optimize as $\alpha$. The bi-level optimization method adopted in [1][2] requires that the lower-level optimization objective is a explicit function of the upper task weights $\alpha$, $L_{train}(\theta,\alpha)$, so that they can utilize the implicit function theorem[1] or first-order approximation[2]. However, our work needs to manipulate $\alpha$ on the gradient of each module, and the lower optimization objective is a implicit function of $\alpha$. If we want to utilize the method in [1][2] to conduct upper optimization, we have to obtain the explicit form of $L_{train}(\theta,\alpha)$ through intractable integration, which is impossible in practice. Therefore, we propose to conduct the upper optimization through unrolling the lower steps and adopting the best-response approximation for efficiency, which is novel compared to [1][2].
>
>     + [4] is an auxiliary learning method that decomposes the auxiliary gradient into the helpful, neutral and harmful component. However, its decomposition lacks our automated module-level consideration:
>
>         + Module-level consideration: [4] decomposes the auxiliary gradient into the helpful, neutral and harmful, three components in the data sample space, and the gradient is still considered in the whole model level, and the subspace is constructed by different samples instead of modules.  Therefore, they do not consider the specific module-level impact.
>
>         + Automation: In [4], the coefficients before the helpful, neutral and harmful components still need tuning, but our proposed method is automatic in discovering the proper auxiliary importance $\alpha$.
>
>     + [3] is a gradient manipulation method for multitask learning, and the differences between our work and this work lie in the following aspects:
>
>         + Motivation: [3] aims at tackling the conflicts among the gradients from multiple tasks, so that the model can achieve acceptable performance on all the tasks. However, our setting focuses on the primary task, and even to improve the primary performance by sacrificing the performance of the harmful and unrelated auxiliary losses.
>
>         + Automation: The method to consider the module-level impact in [3] is a heuristic one for multi-task learning, which cannot guarantee the optimal performance on the primary task. In contrast, our method automatically learns the optimal module-level importance for the primary task, with the guidance of the primary developing dataset. Moreover, our method can be even applied to more generalized auxiliary losses(like L2 or other kinds of regularization) besides the auxiliary task loss thanks to its automation.
>
> Out of above considerations, we think that our work has its clear innovations compared to the existing works.
>
> [1] Should we be pre-training? an argument for end-task aware training as an alternative.
>
> [2] Auxiliary learning by implicit differentiation.
>
> [3] Gradient Vaccine: Investigating and Improving Multi-task Optimization in Massively Multilingual Models.
>
> [4] Auxiliary task update decomposition: the good, the bad and the neutral.

---

> > ### Comment · Reviewer_xStN · 2022-08-07
> > **Rebuttal Response**
> >
> > Apologies for the delay in responding to this.
> >
> > **Novelty of Work**
> >
> > *about [1][2]* - my critique in this respect was not about the specifics of the meta-learning algorithm but rather its particular usage in this case. You are correct in noting that the IFT approach presented in [1,2] cannot be directly used in this setting - but again, that contribution / technique was introduced in  [citation 26 in your paper : https://openreview.net/pdf?id=hfU7Ka5cfrC].
> > My argument here is that meta-learning adaptive whether the way it's done in [1, 2, or https://openreview.net/pdf?id=hfU7Ka5cfrC] has been quite explored.
> >
> > *about [4]* - The method presented here projects the gradients into a subspace of dimensionality k.  This is what I refer to as modularity. The operations conducted within the subspace (helpful, neutral, hurtful) are then what mirror the classical gradient alignment techniques.
> >
> >
> > **Model Size Ablation**
> > In relation to model size, I meant an ablation over all the baselines methods (or at least the strongest other baseline outside of your method) and not just for your method only to see how your method's performance changes with model scale - it is folklore that auxiliary learning techiques exhibit different performance at different model scales. I do not expect you to run this experiment given the short time available - but I think it would be a good thing to have in the later versions of the paper.
> >
> >
> > I have made clear that I think the novelty of the approach is limited. However, I recognize the effort that the authors put into responding to my concerns and engaging with related work.  I update my score from 3->5 to reflect that I think this is a technical details are correct but the novelty is limited in my humble opinion.

---

> > > ### Author Response · Authors · 2022-08-08
> > > **Reply to the Rebuttal Response**
> > >
> > > We thank the reviewer for the efforts in carefully reading our rebuttal and giving new constructive suggestions. We also appreciate that the reviewer acknowledges our work through the rebuttal. Furthermore, we would also like to present some additional remarks on the novelty of this work.
> > >
> > > The detailed discussions with the reviewer make the position of this work clearer. The reviewer points out that the main techniques of our work are inspired by a combination of multiple works from multiple domains,  including multi-task learning [3], learning with auxiliary losses[1][2], bi-level optimization[citation 26], etc. However, we’d like to clarify that the combination is non-trivial,  as we provide a novel perspective, i.e., considering the module-level impact of each auxiliary loss in the generic setting of auxiliary learning. Under this novel perspective, we make the following contributions:
> > >
> > > + We propose a generic, plug-in-like, and effective framework to learn the auxiliary importance, by integrating techniques from multiple domains into a unified solution. Note that the work [citation 26] only considers hyper-parameter optimization, like learning rate, without giving any clues that we should assign each (auxiliary loss, module) pair an individual weight.
> > > + We provide extensive experiments with insightful analyses of various auxiliary learning scenarios. We believe that these insightful experiments and analyses will have important influences on various settings when researchers want to improve their target task when utilizing auxiliary losses. Also, the generic form of our method makes it a generic and powerful tool to be directly applied to various scenarios.
> > >
> > > Although we appreciate the reviewer’s opinion from a purely technical perspective to judge this work,  for the new and potentially important scenario of module-aware auxiliary loss optimization investigated in our work, non-trivially taking the advantage of works from multiple domains to form a generic and effective solution is also meaningful in our humble opinion.
> > >
> > > Additionally, as for the model-size ablation about the folklore that “auxiliary learning techniques exhibit different performance at different model scales”, it seems an interesting problem worth investigating. Although it may be out of the focus of this work, we will add the extra experiments in our final version of the supplementary file.
> > >
> > > Thank you for your constructive comments again!

---

> ### Author Response · Authors · 2022-08-01
> **Reply to the Question part of Reviewer xStN**
>
> Thank the reviewer for the careful questions.
>
> 2. **Reply to Question 1**:
>
>     **Bolded numbers**: Thanks for the careful reading. The bolded numbers are the method with the highest performance, and we will make it clearer in our paper.
>
> 3. **Reply to Question 2**:
>
>     **Experiments**: All the experiments in Table 1,2 are conducted with Block-L. We will more clearly state it in our paper.
>
> 4. **Reply to Question 3**:
>
>     **Recommendation for using Parameter/Block/Model-L**: When the labels for the primary task are enough to afford a large developing dataset, the Parameter-L split can be conducted. Block-L compared to Model-L considers the loss conflicts in different modules but does not introduce too many learned weights(usually 10~100), which is more worth trying as the experimental results shown in our paper.
>
> 5. **Reply to Question 4**:
>
>     **Results change with model size and dev set size**: We present the learned importance changes with model size for the Amazon Recommendation in Figure 5 in the paper. The recommendation RMSE changes are shown in the following table, where larger model seems to obtain comparatively better results with our adaptive module-wise L2 regularizer. We also change the size of dev set in the Pet experiment and keep other conditions the same, and the results are shown in the following table. We find that larger size in the dev set will improve the performance of Parameter-L, but the performance of Module-L is not significantly improved. This also indicates that more parameters for the upper optimization require more data in the dev set. When the number of data samples reaches more than needed, the performance will not substantially improve.
>
>     |  Model   | hidden8  | hidden16 | hidden32 | hidden64 | hidden128 |
>     |  ----  | ----  | ----|----  | ----  | ----|
>     | Amazon RMSE  |$1.1540_{0.0008}$ | $1.0921_{0.0014}$|$1.0882_{0.0018}$|$1.0917_{0.0004}$| $1.0892_{0.0015}$|
>
>     |  dev size(# of the whole dataset)   | 1%  | 5% | 10% | 20% |
>     |  ----  | ----  | ----|----  | ----  |
>     | Pet acc(%) block-L  |$69.61_{0.93}$ | $69.27_{0.51}$|$69.52_{0.63}$| $69.38_{0.49}$|
>     | Pet acc(%) Parameter-L  |$66.14_{0.87}$ | $68.32_{1.09}$|$69.93_{0.53}$| $69.45_{0.87}$|
>
>
> 6. **Reply to Question 5**:
>
>     **Learning rate tuning**: In the experiments in the paper, we use the HPO-tune baseline to search for a proper learning rate and use this learning rate for all the methods for fair comparison, since we think that the auxiliary learning methods only balance the weights and the learning rate should be kept the same for all the methods. In our new experiments on the NYUv2 dataset and cifar100-20 dataset, we conduct random search for the learning rate of each method as the reviewer suggested. Specifically, in the NYUv2 dataset, we adopt the EfficientNet as the backbone, and regard the segmentation as the primary task with depth prediction and normal prediction as auxiliary tasks. In the cifar100-20 dataset, we treated each of the 20 ‘coarse’ classes as one task, where each coarse class contains 5 classes. We regard the 'people' classification as the primary task, and the 0-9 'coarse' classes as auxiliary tasks, totally 11 tasks. The ResNet18 is adopted as the backbone for cifar100-20. The experimental results in the following table show that our method also outperforms the baselines. Note that Equal is a new baseline that assigns all losses with weight "1.0".
>
>     |  NYUv2   | SLL  | Equal | HPO-tune | Uncert | GCS| AuxL | Ours |
>     |  ----  | ----  | ----|----  | ----  | ----|----  | ----  |
>     | m-IOU(%)  |$32.61_{0.21}$  | $33.15_{0.15}$| $33.09_{0.25}$| $32.93_{0.32}$| $32.18_{0.34}$| $33.06_{0.14}$| $\textbf{33.88}_{0.23}$|
>     | pixel-acc(%) | $68.42_{0.62}$ |$68.11_{0.24}$|$68.52_{0.52}$| $68.74_{0.62}$| $ 67.58_{0.80}$|$68.36_{0.53}$| $\textbf{69.17}_{0.44}$|
>
>     |  CIFAR100-20   | SLL  | Equal | HPO-tune | Uncert | GCS| AuxL | Ours |
>     |  ----  | ----  | ----|----  | ----  | ----|----  | ----  |
>     | acc(%)  |$49.73_{0.61}$  | $51.27_{0.61}$|$51.75_{0.72}$|$48.40_{0.72}$| $51.80_{0.40}$|  $51.53_{0.23}$| $\textbf{53.87}_{0.46}$|
>
>
> 7. **Reply to the note**:
>
>     **Weights evolution**: We add the weights evolution in Section 3 in the appendix. Thanks for your suggestion.

---

> ### Author Response · Authors · 2022-08-01
> **Reply to the Limitation part of Reviewer xStN**
>
> 8. **Reply to limitation 1**:
>
>     **The additional hyper-parameters**: For the granularity, we regard each block as a module and achieve consistent performance improvement under various settings. M in our experiments is fixed to 3 as suggested in [5], which also shows good generalization ability to various scenarios in our experiments. From this perspective, we think our method will not bring heavy burden for HPO.
>
> 9. **Reply to limitation 2**:
>
>     **Computational Cost**: Regarding the time of joint optimization of multiple losses with fixed weights as unit "1", denoting the number of losses as N, and considering that time cost differences of the fixed training and our method mainly come from the backward process, the backward time complexity of our method is as follows:
>
>     In one lower-upper loop, the model will conduct T times of lower optimization and 1 upper optimization. The cost time for fixed weight training will be T. Our method will have TN backward in the lower optimization. In the upper optimization, our method requires (N+M+1) backward for the Jacobi calculation, where M is the looking-back step, so it needs total (TN+N+M+1) backwards, which results in O((N+M+TN+1)/T) complexity compared to the unit. Since M in the experiments is fixed to 3, our method needs about O(N+N/T) complexity.
>
> [5] Lorraine, J. , Vicol, P. , & Duvenaud, D. . (2019). Optimizing Millions of Hyperparameters by Implicit Differentiation.

---

### Official Review · Reviewer_94wP · 2022-07-10

**Rating:** 5
**Confidence:** 5
**Soundness:** 3 good
**Presentation:** 3 good
**Contribution:** 2 fair

**Summary:**

This paper proposes an auxiliary learning framework that updates network parameters in a module level via learnable module-level auxiliary importance. The proposed framework is updated with bi-level optimisation, a standard way to perform auxiliary learning with meta learning.
As such it contains two stages: the inner loop: it updates the network parameter by reweighting gradients of each auxiliary loss on each network module via the fixed auxiliary importance mask on the training data; the outer loop: it updates the auxiliary importance mask by the performance of primary task with M inner rollout steps, which uses best-response approximation (assuming network weightings are converged in inner loop) to reduce training time and memory on developing data (a small dataset to split from validation dataset). The framework eventually stores the best network which achieves the optimal validation performance.

The paper evaluates on multiple small-scale image classification datasets and user rating prediction datasets for regression with multiple network design and with supervised and semi-supervised learning settings. The auxiliary tasks are specifically designed for each dataset, e.g. using available attribute information as auxiliary tasks for fine-grained bird species classification, or using hand-crafted auxiliary tasks like rotation prediction for image classification datasets without attribute information. The proposed framework was compared to multiple well-known auxiliary learning methods, and was shown to improve the performance with a non-trivial margin.


**Questions:**

1. Is the auxiliary importance mask bounded or unbounded?

2. Could you provide the experiments with 3 or more auxiliary tasks, in the standard multi-task/auxiliary task literature, e.g. in NYUv2, CityScapes, CIFAR-100/20 classification, as shown in [1, 2]?

[1] Liu et al. Auto-Lambda: Disentangling Dynamic Task Relationships, 2022
[2] Yu et al. Gradient Surgery for Multi-Task Learning, 2020

3. Could you provide the experiments with Equal weighting, so it shows the benefit of auxiliary learning method with the same training data?

4. Minor comment: The dataset chosen in Section 4.4 looks a bit suspicious, which includes two settings having the large performance gap.


**Strengths And Weaknesses:**

I found the paper is very easy to follow with very clean writing. The visualisation and explanation of auxiliary importance is also interesting to read.

However, I think the design of experiments is a bit insufficient.
1) the number of auxiliary tasks is only limited to two, most of the time only has one auxiliary task.

2) the benefits on auxiliary tasks are not well reflected in the experiments. E.g. L267-269 claims the related tasks will bring more benefits, but Table 1 shows training with auxiliary tasks simply not brought any performance improvement, on CUB dataset.

3) I wouldn’t call L2 regularisation is an auxiliary task, though it still might fits in a general definition…

4) missing simple baseline: Equal weighting as multi-task learning, just to see the benefit of the proposed/baseline auxiliary methods with exact same training data.

---

> ### Author Response · Authors · 2022-08-01
> **Reply to Reviewer 94wP**
>
> Thank the reviewer for the constrcutive comments and suggestions. We address your concerns point by point as follows.
> 1. **Reply to Weakness 1 & Question 2**:
>
>     **Additional experiments**: Thank the reviewer for the suggestions. We add the NYUv2 and CIFAR-100/20 experiments. In the NYUv2 dataset, we adopt the EfficientNet as backbone, and in the CIFAR-100/20, we adopt ResNet18 as the backbone. Note that we choose "people" classification as the primary task, and use the 0-9 coarse class as the auxiliary tasks. Totally, there are 11 tasks trained together. The results are shown in the following table, which further validate the effectiveness of our proposed method.
>
>     |  NYUv2   | SLL  | Equal | HPO-tune | Uncert | GCS| AuxL | Ours |
>     |  ----  | ----  | ----|----  | ----  | ----|----  | ----  |
>     | m-IOU(%)  |$32.61_{0.21}$  | $33.15_{0.15}$| $33.09_{0.25}$| $32.93_{0.32}$| $32.18_{0.34}$| $33.06_{0.14}$| $\textbf{33.88}_{0.23}$|
>     | pixel-acc(%) | $68.42_{0.62}$ |$68.11_{0.24}$|$68.52_{0.52}$| $68.74_{0.62}$| $ 67.58_{0.80}$|$68.36_{0.53}$| $\textbf{69.17}_{0.44}$|
>
>     |  CIFAR100-20   | SLL  | Equal | HPO-tune | Uncert | GCS| AuxL | Ours |
>     |  ----  | ----  | ----|----  | ----  | ----|----  | ----  |
>     | acc(%)  |$49.73_{0.61}$  | $51.27_{0.61}$|$51.75_{0.72}$|$48.40_{0.72}$| $51.80_{0.40}$|  $51.53_{0.23}$| $\textbf{53.87}_{0.46}$|
>
> 2. **Reply to Weakness 2**:
>
>     **The benefits of auxiliary tasks on CUB**: Line267-269 refers to the CUB-semi results in Table 2 instead of Table 1, where the methods with auxiliary losses show better performance than only the target loss. We will make it clearer in the future version. In the CUB dataset, we found that when we use all data for the target task, the auxiliary tasks are not beneficial. When the labels for the target task are inadequate, the auxiliary tasks become beneficial.
>
> 3. **Reply to Weakness 3**:
>
>     **L2 regularization**: Considering that the losses of auxiliary tasks in fact impose additional regularization on the model parameters, we would like to investigate whether our method can handle more generalized auxiliary losses like L2 regularization. Additionally, auxiliary losses are not always from auxiliary tasks, like the most widely used L2 regularization in recommendation or even disentangled losses in [1][2]. We expect our method can handle various auxiliary losses, so we add the two experiments of recommendation with L2 regularization to prove that our method is generic.
>
> 4. **Reply to Weakness 4 & Question 3**:
>
>     **Equal weighting baseline**: Thanks for your suggestion. We add the experiments of Equal weighting, and the results are shown in the following table. Note that in the recommendation task the Equal baseline achieves bad performance, because L2 regularization coefficient "1.0" is too large to obtain promising recommendation results.
>     |  Dataset   | CUB  | CUB-semi | Pet | Pet-semi | CIFAR10 | CIFAR100 | Beauty(RMSE) | MovieLens(RMSE)|
>     |  ----  | ----  | ----|----  | ----  | ----|----  | ----| ----|
>     | SLL(single loss learning)  |$77.29_{0.63}$  | $48.15_{0.88}$|$61.14_{1.03}$|$31.81_{1.17}$| $71.60_{0.25}$|  $46.52_{0.42}$| $1.1008_{0.0006}$ |$0.9068_{0.0028}$|
>     | Equal  |$71.86_{0.87}$  | $50.54_{0.55}$|$67.52_{0.58}$|$49.07_{0.94}$| $70.60_{0.30}$|  $45.57_{0.25}$| $2.4512_{0.0013}$| $2.2567_{0.0052}$
>     | ours  |$\textbf{77.30}_{0.17}$  | $\textbf{52.36}_{0.64}$|$\textbf{69.61}_{0.93}$|$\textbf{51.42}_{0.31}$| $\textbf{72.24}_{0.27}$|  $\textbf{47.57}_{0.40}$| $\textbf{1.0873}_{0.0017}$ | $\textbf{0.9045}_{0.0013}$|
> 5. **Reply to Question 1**:
>
>     **The importance masks are bounded or not** : The mask is bounded by zero. We will make sure that the importance mask non-negative with relu function, and we will make it clearer in our paper. Thanks for your careful question.
>
>
> 6. **Reply to Question 4**:
>
>     **The dataset chosen in Section 4.4**: We rerun our experiment and achieve the same results. The performance gap may be explained by the following reasons after our analysis. It's worth noticing that as we clarify in the supplementary file, the experiments on the CUB dataset are based on a pretrained ResNet18, and we conduct finetuning on this dataset. Therefore, even conducting the parameter-level importance weighting, the model still achieves good enough performance without overfitting thanks to the knowledge of the pretraining model. However, in the Pet dataset, we train from scratch, all the knowledege of the model comes from the learned tasks, small developing dataset will easily result in overfitting problem in the parameter-level setting as also pointed out in [3]. Therefore, the results are reasonable. Thank you for careful reading and interesting phenomenon discoveries in our paper.
>
> [1] Learning Disentangled Representations for Recommendation.
>
> [2] Disentangled Graph Collaborative Filtering.
>
> [3] Optimizing Millions of Hyperparameters by Implicit Differentiation.

---

> ### Author Response · Authors · 2022-08-08
> **Gentle Reminder to the Rebuttal**
>
> Dear Reviewer 94wP,
>
> As the author-reviewer discussion period is about to end, we hope that our responses have provided adequate information to address the reviewers' concerns. We'd like to make sure that we haven't missed anything important. Please feel free to let us know if there are any further comments or suggestions. We are happy to answer further questions if necessary.
>
> Best

---

> > ### Comment · Reviewer_94wP · 2022-08-08
> > **Response to Rebuttal**
> >
> > Thank the authors for their rebuttal.
> >
> > Most of my concerns have now been resolved. I would like to keep my original rating as my final rating.

---

### Official Review · Reviewer_NPMU · 2022-07-10

**Rating:** 6
**Confidence:** 3
**Soundness:** 3 good
**Presentation:** 3 good
**Contribution:** 3 good

**Summary:**

This paper focuses on improving the effectiveness of leveraging auxiliary objectives for training deep neural network models. The authors propose a learning algorithm that jointly optimizes 1) the weight of each auxiliary loss on different parts (modules) of the model and 2) model parameters according to the weighted sum of the loss terms. Experiments are conducted on image classification and user rating prediction tasks to verify the efficacy of the proposed approach.

**Questions:**

Please see Weakness.

**Limitations:**

The authors address the limitation in Conclusion. No obvious potential negative social impact is observed.

**Strengths And Weaknesses:**

*Strengths*
1. The paper is well-written. The idea is sound, and the technical details are clear.
2. The proposed concept may have important contributions to the community. Indeed the impact of auxiliary training losses on different parts of the model is not well-understood yet.
2. The paper provides some interesting observations in the experiments, especially the loss weight visualizations in Section 4.3.
*Weakness*
1. The experiments are conducted on a rather small scale (). The proposed approach could be more promising if the authors apply the algorithm to 1) more SOTA networks, e.g., VIT or EffiecientNet, and 2) more large-scale tasks, e.g., semi-supervised object detection.
2. In Line 294, the authors use the descriptions, "That is to say, the gradient from the auxiliary losses may guide θ to a better descent direction for the primary task in the early stages, but the gradient will show little importance when the better direction is found" to explain those cases where the loss weight is learned to be 0. It would be great to visualize the change of the loss weight over time (training loops) to support the authors' arguments here.
3. The authors should report the computational loss in the paper since it is the limitation of the proposed approach.

---

> ### Author Response · Authors · 2022-08-01
> **Reply to Reviewer NPMU**
>
> Thank you for your careful comments and constructive suggestions. We address your concerns point by point as follows.
>
> 1. **Reply to Weakness in 4**:
>
>     **Additional experiments**: We conduct additional experiments on the NYUv2 dataset and cifar100-20 dataset, which are widely used in the auxiliary learning literature[1][2]. In the NYUv2 dataset, we adopt the EfficientNet as the backbone as you suggest, and regard the segmentation as the primary task with depth prediction and normal prediction as auxiliary tasks. In the cifar100-20 dataset, we treated each of the 20 ‘coarse’ classes as one task, where each coarse class contains 5 classes. We regard the "people" classification("baby", "boy", "girl", "man", "woman") as the primary task, and the 0-9 coarse class classification as the auxiliary tasks, totally 11 tasks. The ResNet18 is adopted as the backbone for the CIFAR100-20. The experimental results are shown in the following table, which further supports the effectiveness of our method. Note that we also add the Equal baseline(all losses have the same weight "1.0"), which can directly help to judge whether using auxiliary tasks can be beneficial to the primary task.
>
>     |  NYUv2   | SLL  | Equal | HPO-tune | Uncert | GCS| AuxL | Ours |
>     |  ----  | ----  | ----|----  | ----  | ----|----  | ----  |
>     | m-IOU(%)  |$32.61_{0.21}$  | $33.15_{0.15}$| $33.09_{0.25}$| $32.93_{0.32}$| $32.18_{0.34}$| $33.06_{0.14}$| $\textbf{33.88}_{0.23}$|
>     | pixel-acc(%) | $68.42_{0.62}$ |$68.11_{0.24}$|$68.52_{0.52}$| $68.74_{0.62}$| $ 67.58_{0.80}$|$68.36_{0.53}$| $\textbf{69.17}_{0.44}$|
>
>     |  CIFAR100-20   | SLL  | Equal | HPO-tune | Uncert | GCS| AuxL | Ours |
>     |  ----  | ----  | ----|----  | ----  | ----|----  | ----  |
>     | acc(%)  |$49.73_{0.61}$  | $51.27_{0.61}$|$51.75_{0.72}$|$48.40_{0.72}$| $51.80_{0.40}$|  $51.53_{0.23}$| $\textbf{53.87}_{0.46}$|
>
> 2. **Reply to Weakness in 5**:
>
>     **Change of weights during training**: We add the change of the importance weights over time into the section 3 in the appendix. Thanks for your suggestions.
>
> 3. **Reply to Weakness in 6**:
>
>     **The computational cost analysis**: The computational loss is as follows and we add it to the section 4 in the appendix. Regarding the time of joint optimization of multiple losses with fixed weights as unit "1", and considering that the time cost differences mainly come from the backward process, the backward time complexity of different methods is as follows:
>
>     + SLL(single loss learning): O(1).
>     + HPO-tune: O(R), in our method we use random search for the weights, R is the maximal trial number of the random search process.
>     + uncert: O(1)
>     + GCS(gradient cosine similarity): O(N), N is the number of losses. Since GCS will calculate the gradient for each loss, it will need N times of backward when updating the model parameters.
>     + AuxL: O(1+5/T). T is the number of lower steps in one lower-upper loop. This method is a bi-level approach, we compare its time complexity with the fixed weight training method in one lower-upper loop.  In one loop, the model will conduct T times of lower optimization and 1 upper optimization. The cost time for fixed weight training will be O(T). AuxL will have the additional uppper optimization which requires (M+2) for the Jacobi calculation where M is the looking-back steps, so it needs total (T+M+2) backwards, which results in O((M+T+2)/T) complexity compared to the fixed weight training method. Since M in our experiments is fixed to 3, AuxL needs about O(1+5/T) complexity.
>     + MAOAL: O(N+N/T). Compared to AuxL, our method needs to calculate the gradient for each loss in the lower optimization, which requires TN backward. In the upper optimization, our method has N-1 more backward than AuxL to calculate the gradient of each loss. Therefore, the time complexity is O((TN+N+M+1)/T) = O(N+N/T) in our experiment.
>
> [1] Navon, A., Achituve, I., Maron, H., Chechik, G., and Fetaya, E. Auxiliary learning by implicit differentiation. In 9th International Conference on Learning Representations, ICLR 2021, Virtual Event, Austria, May 3-7, 2021, 2021.
>
> [2] Liu et al. Liu, S. , James, S. , Davison, A. J. , & Johns, E. Auto-lambda: disentangling dynamic task relationships.

---

> > ### Comment · Reviewer_NPMU · 2022-08-08
> > **Feedback to author responses**
> >
> > Thank you for the responses. I believe that those address my concerns raised in the review. Nice work!

---

> ### Author Response · Authors · 2022-08-08
> **Gentle Reminder to the Rebuttal**
>
> Dear Reviewer NPMU,
>
> As the author-reviewer discussion period is about to end, we hope that our responses have provided adequate information to address the reviewers' concerns. We'd like to make sure that we haven't missed anything important. Please feel free to let us know if there are any further comments or suggestions. We are happy to answer further questions if necessary.
>
> Best

---

### Official Review · Reviewer_s6Tb · 2022-07-11

**Rating:** 8
**Confidence:** 5
**Soundness:** 4 excellent
**Presentation:** 4 excellent
**Contribution:** 4 excellent

**Summary:**

This paper proposes a module-aware optimization method for a commonly adopted learning paradigm, learning with auxiliary losses. Inspired by multi-task learning, the authors point out that different auxiliary losses will contribute differently to different modules of the model for improving the primary task. To obtain the optimal importance of each auxiliary loss to each module, the authors design an efficient bi-level optimization based method. Extensive experiments on several datasets and tasks show that the proposed method for auxiliary learning is effective and generic. Additionally, the authors conduct comprehensive ablations from different aspects, which offers insights for auxiliary learning in various scenarios.

**Questions:**

1. The authors propose to utilize the validation set to prevent overfitting the developing set and the training set. I’d like to know whether the validation set for the proposed method and the baselines are the same, or the validation set for the baselines contain the developing set?
2. I’d also like to see the time complexity of different baselines and this method.


**Limitations:**

The authors point that the time complexity is the limitation of this method, but it needs more discussing as above.

**Strengths And Weaknesses:**

Strengths:
1.	This paper is well-motivated. The module-aware optimization is important to addressing the module-level conflicts between the primary task and the auxiliary losses.
2.	The related work and the contributions of this paper are sufficiently discussed. It is clearly discussed the differences between existing auxiliary learning, multi-task learning methods and the proposed method. The necessity of this work is obvious.
3.	The proposed method is novel and efficient. The formulated bi-level optimization problem requires to make the gradient of the model parameterized, which is usually memory and time consuming to conduct the upper optimization. With the hyper-gradient derivation, the authors adopt the efficient best-response approximation, which gives a nice solution.
4.	The experiments with different datasets, backbones and the ablations are convincing. The analysis of the whole experiments is quite insightful, and some results are quite interesting, like the learned module-level weights changing with the complexity of the modules.
5.	This paper is well-written and easy to follow. The proposed method is a general method and can be easily applied to various auxiliary learning scenario, which could be a useful plug-in for many tasks.

Weaknesses:
1.	The weakness of this paper lies in that although the proposed method is efficient compared to the directly unrolling method. The time complexity of both the baselines and this method should also be discussed so that readers can better balance these methods.
2.	Although the module choice effect is shown in the experiments, there needs more analysis and instructions for readers to decide when to conduct finer-grained module split or must we conduct parameter-level module split with collecting more data in the developing set?

---

> ### Author Response · Authors · 2022-08-01
> **Reply to Reviewer s6Tb**
>
> Thank the reviewer for constructive comments and suggestions. We address your concerns point by point as follows.
>
> 1. **Reply to Weakness 1& Question2 & Limitation**:
>     **Time complexity analysis of different methods**:
>     Regarding the time of joint optimization of multiple losses with fixed weights as unit "1", and considering that the computational differences of different methods mainly come from the backward process, the backward time complexity of different methods is as follows:
>
>     + SLL(single loss learning): O(1).
>     + HPO-tune: O(R), in our method we use random search for the weights, R is the maximal trial number of the random search process.
>     + uncert: O(1)
>     + GCS(gradient cosine similarity): O(N), N is the number of losses. Since GCS will calculate the gradient for each loss, it will need N  backward when updating the model parameters.
>     + AuxL: O(1+5/T). This method is a bi-level approach, we compare its time complexity with the fixed weight training method in one lower-upper loop.  In one loop, the model will conduct T times lower optimization and 1 upper optimization. The cost time for fixed weight training will be O(T). AuxL will have the additional upper optimization which requires (M+2) for the Jacobi calculation where M is the looking-back steps, so it needs total (T+M+2) backwards, which results in O((M+T+2)/T) complexity compared to the unit. Since M in our experiments is fixed to 3, AuxL needs about O(1+5/T) complexity.
>     + MAOAL: O(N+N/T). Compared to AuxL, our method needs to calculate the gradient for each loss in the lower optimization, which requires TN backward. In the upper optimization, our method has (N-1) more backward than AuxL to calculate the gradient of each loss. Therefore, the time complexity is O((TN+N+M+1)/T) = O(N+N/T) in our experiment.
>
> 2. **Reply to Weakness 2**:
>
>     **The module split principle**: The finer-grained(like parameter-level) split can be conducted if there are more data samples in the developing dataset as shown in both our experiments and [1]. In general, when we want to achieve better performance for our target task with auxiliary losses, we possibly lack data samples for the target task(If we have abundant data for the target task, training on these data can achieve good enough performance and auxiliary losses may not be so required[2][3][4]). Obtaining large enough developing datasets for auxiliary learning is usually difficult. Therefore, in the setting of auxiliary learning, conducting module-level optimization for auxiliary losses as shown in our paper is a proper choice.
>
> 3. **Reply to Question 2**
>
>     **The validation dataset**: The validation datasets for all the methods are the same. The developing dataset is only used for optimizing the upper-level weights for the bi-level methods, and used for training for other baselines.
>
>
> [1] Lorraine, J. , Vicol, P. , & Duvenaud, D. . (2019). Optimizing Millions of Hyperparameters by Implicit Differentiation.
>
> [2] Lin, X., Baweja, H. S., Kantor, G., and Held, D. Adaptive auxiliary task weighting for reinforcement learning. In NeurIPS 2019, December 8-14, 2019, Vancouver, BC, Canada, pp. 4773–4784, 2019.
>
> [3] Navon, A., Achituve, I., Maron, H., Chechik, G., and Fetaya, E. Auxiliary learning by implicit differentiation. In 9th International Conference on Learning Representations, ICLR 2021, Virtual Event, Austria, May 3-7, 2021, 2021.
>
> [4] Du, Y., Czarnecki, W. M., Jayakumar, S. M., Pascanu, R., and Lakshminarayanan, B. Adapting auxiliary losses using gradient similarity. ICLR, 2018.

---

> ### Author Response · Authors · 2022-08-08
> **Gentle Reminder to the Rebuttal**
>
> Dear Reviewer s6Tb,
>
> As the author-reviewer discussion period is about to end, we hope that our responses have provided adequate information to address the reviewers' concerns. We'd like to make sure that we haven't missed anything important. Please feel free to let us know if there are any further comments or suggestions. We are happy to answer further questions if necessary.
>
> Best

---

### Author Response · Authors · 2022-08-07
**Rebuttal Follow-up**

Dear reviewers,

Thank all the reviewers for the constructive comments and questions. As the author-reviewer discussion period is about to end, we hope that our responses have provided adequate information to address the reviewers' concerns. We'd like to make sure that we haven't missed anything important. Please feel free to let us know if there are any further comments or suggestions. We are happy to answer further questions if necessary.

Best

---

### Meta-Review · Area_Chair_2Hrg · 2022-08-23

**Recommendation:** Accept
**Confidence:** Less certain

**Metareview:**

The paper investigates module-aware optimization for auxiliary learning, which adjusts the impact of the auxiliary tasks (through their losses) at the module-level instead of treating the model as a whole. The motivation is based on the observation that a certain auxiliary loss may be beneficial for optimizing specific modules in a model but harmful to others at the same time.  A bi-level formulation is employed as a natural solution, where the inner loop optimizes the model parameters and the outer loop optimizes the module-level auxiliary importance. Experiments conducted using different auxiliary losses on diverse datasets show the advantage of the proposed approach over existing baselines.

Overall, the paper is well written and the problem is clearly motivated. The solution is technically sound and the experimental results show the effectiveness of the proposed module-aware optimization approach. The authors have conducted additional experiments based on the reviewers’ suggestions, which further enhanced the evaluation side of the paper. On the other hand, the overall novelty does not appear to be very strong. Treating the importance of each auxiliary loss to each module as  hyper-parameters and then using a bi-level optimization to optimize both the model parameters and the importance parameters is a very straightforward idea. Reviewers also pointed out some important related works on learning adaptive weights over auxiliary tasks via bi-level optimization, adapting auxiliary tasks weights on a per-module basis,  and layer-wise weight adaptation. The authors are encouraged to reference these important related works and highlight the core technical differences.


**Award:**

No

---

### Decision · Program_Chairs · 2022-09-14

Accept